# Clinical Significance of Serum Membrane-Bound Mucin-2 Levels in Breast Cancer

**DOI:** 10.3390/biom9020040

**Published:** 2019-01-24

**Authors:** Suleyman Bademler, Alisan Zirtiloglu, Murat Sari, Muhammed Zubeyr Ucuncu, Elif Bilgin Dogru, Senem Karabulut

**Affiliations:** 1Department of Surgery, Institute of Oncology, Istanbul University, 34093 Istanbul, Turkey; sbademler@gmail.com; 2Department of Medical Oncology, Bakırkoy Dr Sadi Konuk Education and Research Hospital, 34147 Istanbul, Turkey; 3Department of Medical Oncology, Institute of Oncology, Istanbul University, 34093 Istanbul, Turkey; drmuratsari@gmail.com; 4Department of Health Science Institute, Istanbul Gelisim University, 34310 Istanbul, Turkey; muhammeducuncu@gmail.com; 5Department of Basic Oncology, Institute of Oncology, Istanbul University, 34093 Istanbul, Turkey; elifbilgin85@gmail.com; 6Department of Medical Oncology, Institute of Oncology, Istanbul University, 34093 Istanbul, Turkey; drsenemkarabulut@gmail.com

**Keywords:** breast cancer, MUC2, serum, diagnostic

## Abstract

This study was conducted to investigate the serum levels of membrane-bound mucin 2 (MUC2) in breast cancer (BC) patients and the relationship with tumour progression and known prognostic parameters. We enrolled 127 female patients with histopathologically diagnosed BC who did not receive chemotherapy (CT) or radiotherapy. Serum *MUC*2 levels were measured by the enzyme-linked immunosorbent assay (ELISA) method and compared with those of 40 age and sex-matched healthy controls. Median age of diagnosis was 50 (range: 26–78). Twenty-eight (22%) patients were metastatic and the most frequent site of metastasis was bone (*n* = 17, 61%). The median serum *MUC*2 level of BC patients was significantly higher than that of the controls (198 vs. 54 ng/mL, *p* < 0.001). There was no significant difference between patients and controls according to known disease-related clinicopathological or laboratory parameters (*p* > 0.05). Serum *MUC*2 levels were not associated with survival (*p* = 0.65). Although serum *MUC*2 levels might have a diagnostic role, their predictive and prognostic role in survival in BC patients was not detected. Serum levels of *MUC*2 should be investigated for diagnostic or screening purposes on a larger scale.

## 1. Introduction

For females in both developed and developing countries, breast cancer (BC) has the second highest mortality rate after lung cancer. More than 1.3 million individuals are diagnosed with BC each year, and the mortality rate is 60% in developing countries [1,2]. Some studies have reported that in American women, the probability of developing BC is 12.3% [3]. Presently, various factors such as genetic predisposition, hormones, lifestyle, and age play an etiological role in this disease [4]. Approximately 7% of patients with BC are diagnosed before age 40 years and the risk for BC increases with age. The most critical factors affecting the survival of patients with BC are the early diagnosis, tumour stage, and age [3,5].

Mucins are high molecular weight glycoproteins, and are sub-classified into two structurally different groups: the secreted and the membrane-bound (transmembrane) mucins [6]. Membrane-bound mucins act as a cell surface receptor for various signal transduction pathways. Secreted mucins form a protective layer and form a barrier against pathogens. Mucins are found in the mammary glands, liver, and in pancreas and kidney secretions, primarily in the respiratory and digestive system [7]. Many studies have shown the altered expression of mucins in different pathological conditions, including cancer. The hypersecretion of mucins in cancers is due to their distinctive pattern of glycosylation, that serves as a binding platform for various growth factors and cytokines and thereby promotes the proliferation and metastasis of malignant cells through several signalling cascades. Each mucin has a specific role in tumour formation; some of the membrane-bound mucin (*MUC*) genes show hyperactivation and others show inactivation in pathological conditions. Membrane-bound mucin-2 (*MUC*2) is a secretory mucin, which is mainly in the gel format and is usually excreted in the digestive and respiratory tract [8,9]. *MUC*2 overexpression is seen in mucinous carcinomas in organs such as the pancreas, breast, and prostate [10,11,12]. Dhanisha et al. suggested that an increased level of mucin in the serum can be used as a potential marker for disease diagnosis and prognosis [6].

In some preclinical studies, *MUC*2 expression was evaluated in breast cancer tissue [12,13,14,15,16,17,18]. However, to the best of our knowledge, the clinical significance of serum *MUC*2 levels in breast cancer has not been previously examined. Therefore, this study was conducted to investigate the serum levels of *MUC*2 in BC patients and the relationship with tumour progression and known prognostic parameters.

## 2. Materials and Methods

### 2.1. Patients

In total, 127 female breast cancer patients admitted to Istanbul University, Institute of Oncology were enrolled in the study. Serum samples were obtained on the first admission before any type of treatment (chemotherapy, radiotherapy, hormonotherapy, or targeted therapy) was given to the patients. Written informed consent was obtained from all patients and healthy controls in order to obtain blood and to conduct *MUC*2 tests for their blood. The pretreatment evaluation included detailed clinical history and physical examination with a series of biochemistry tests, including serum lactate dehydrogenase (LDH) and Cancer Antigen 15-3 (CA15-3) levels. Diagnosis of breast cancer was histologically proven either with tru-cut biopsy of the primary lesion or metastatic site and staged according to the seventh edition of American Joint Committee on Cancer (AJCC). All the patients were treatment-naive for at least 3 months before accrual.

For histological evaluation, tissue sections (2 mm) were deparaffinized and stained using haematoxylin and eosin. Grading of tumours was established according to the modified Bloom-Richardson grading system. Estrogen receptor (ER), progesterone receptor (PR), and HER2 status was evaluated in the sample sections using appropriate antibodies. The immunohistochemical staining was assessed upon visual inspection with an optical microscope and considered as positive if the percentage of cells staining positive was more than 5%. Regarding HER2 evaluation, in the case of 2(+) staining by immunohistochemistry, HER2 gene amplification was analysed by fluorescent in situ hybridization (FISH).

All patients were treated with standard treatment modalities according to NCCN guidelines. Sixty-two patients received adjuvant treatment after mastectomy or lumpectomy. Another sixty-five patients with locally-advanced (*n* = 37) and metastatic disease (*n* = 28) were treated with anthracycline and/or taxane-based chemotherapy (CT) or single-agent capecitabine, and trastuzumab (if HER2 positive) with/without radiotherapy depending on the stage of the disease. Follow-up programs included clinical, laboratory, and radiological assessments performed at 8-week intervals during CT. Response to treatment was evaluated in patients who had metastatic disease or receiving neoadjuvant CT according to clinical examination and the revised RECIST criteria version 1.1. Those with complete or partial responses were considered as CT-responsive. Tumour (T) and Nodal (N) stages were recorded for only non-metastatic patients. While breast cancer has four transcriptional subtypes in the prediction analysis of microarray 50 (PAM50) scheme—luminal A, luminal B, basal-like, and human epidermal growth factor receptor 2 (HER2)-enriched (HER2E)—these subtypes overlap with immunohistochemical (IHC) staining of three protein markers, ER, PR, and HER2, supplemented with in situ hybridization (ISH) of HER2. The HER2E subtype captures some but not all HER2+ tumours [19]. However, we categorized the patients according to molecular subtypes as; Her2-negative luminal (ER positive, HER2 negative), triple positive (ER, PR, and HER2 positive), triple negative (ER, PR, and HER2 negative), and HER2 disease (ER and PR negative, HER2 positive).

For comparison of serum levels of *MUC*2, age and sex-matched 40 healthy women controls were included in the analysis. Our study was found suitable and was approved on 9 March 2018 by the Ethics Committee of Istanbul Faculty of Medicine, Istanbul University with project number 2018/359.

### 2.2. Measurement of Serum MUC2 Levels

Blood samples were obtained from patients with breast cancer and healthy controls by venipuncture and clotted at room temperature. The sera were collected following centrifugation and frozen immediately at –80 °C until analysis. Serum *MUC*2 (USCN, Wuhan, China) levels were determined by the solid-phase sandwich ELISA method. The microtiter plate has been pre-coated with an antibody specific to *MUC*2. Standards and samples were added to the wells with a biotin-conjugated antibody specific to *MUC*2. Next, avidin conjugated to horseradish peroxidase (HRP) was added to each well and incubated. After tetramethylbenzidine (TMB) substrate solution was added, the wells that contain *MUC*2, biotin-conjugated antibody, and enzyme-conjugated avidin exhibited a change in colour. The enzyme substrate reaction was terminated by the addition of sulphuric acid solution and the colour change was measured spectrophotometrically (ChroMate® 4300; Awareness Technology, Inc., Palm City, FL, USA) at a wavelength of 450 nm. The concentration of *MUC*2 in the samples was then determined by comparing the optical density (OD) of the samples to the standard curve.

### 2.3. Statistical Analysis

Continuous variables were categorized using median values as the cut-off point. For assessment of relationships, comparisons between various clinical/laboratory parameters and serum levels of *MUC*2 assays were accomplished using Mann–Whitney U test since serum *MUC*2 had non-normal distribution in all groups. Survival was calculated from the date of the first admission to hospital to the death resulting from any cause or to the last contact with the patient or any family member. Kaplan–Meier method was used for estimation of survival of patient and differences in survivals were assessed by the log-rank statistic. A *p*-value less than 0.05 was accepted as statistically significant. Statistical analysis was carried out using IBM SPSS Statistics for Windows, Version 21.0 (Armonk, NY: IBM Corp. Released 2012, USA).

## 3. Results

Between July 2006 and June 2016, 127 female breast cancer patients were enrolled in the study. Median age of diagnosis was 50 (range: 26-–8) (Table 1). Twenty-eight (22%) patients were metastatic and the most frequent site of metastasis was bone (*n* = 17, 61%). The majority of the patients with non-metastatic disease had node-positive (*n* = 64, 65%) disease and most of them (*n* = 34, 34%) were of the N_1_ stage. The serum *MUC*2 levels showed a strong difference between the patients and the control group (Table 2, Figure 1). Serum *MUC*2 levels were significantly higher than the controls (median level of 198 vs. 54 ng/mL, *p* < 0.001). As a diagnostic test, the sensitivity and specificity of serum *MUC*2 levels was 74.8% and 75%, respectively, if the cut-off value was taken as 97.5 ng/mL (*p* < 0.001) (Figure 2). Positive predictive value was 90.5% and negative predictive value was 48.4% according to the receiver operating characteristic (ROC) analysis.

Table 3 shows the correlation between the serum levels of *MUC*2 and clinicopathological factors. There were no significant correlations between serum *MUC*2 levels and any type of clinic-pathological and laboratory parameters (*p* > 0.05).

### Survival Analysis

The median follow-up time was 65 months (range: 5–97 months). At the end of the observation period, 22 (17%) patients were dead. The mean survival for all patients was 85.0 ± 2.4 months (95% CI = 28.05–43.14) and 1-, 3-, and 5-year overall survival rates were 99% (95% CI = 87–97), 92% (95% CI = 97–101), and 85% (95% CI = 79–92), respectively. Negativity of ER and PR (*p* = 0.03 and *p* = 0.03, respectively), presence of triple-negative histology (*p* = 0.01), higher T stage (*p* = 0.03), elevated CA15-3 (*p* = 0.004), unresponsiveness to chemotherapy (*p* = 0.001), and presence of metastasis (*p* < 0.001) were associated with worse survival (Table 4). However, serum *MUC*2 levels were not associated with survival (*p* = 0.65) (Figure 3).

## 4. Discussion

In our study, we aimed to evaluate serum *MUC*2 levels in patients with breast cancer and age and sex-matched healthy controls. According to our results, serum *MUC*2 levels are higher in the BC patients than the control group. As far as we know, this is the first serum study of *MUC*2 in the literature so far.

*MUC*2 expression is absent in the ductal epithelium of the healthy breast [13]. *MUC*2 is expressed in pre-malignant breast lesions like ductal carcinoma in situ (DCIS), and is upregulated in lobular carcinoma in situ (LCIS). Both membrane-bound (*MUC*1, *MUC*3, *MUC*4) and secreted mucins (*MUC*2, *MUC*5AC, *MUC*5B, *MUC*6) are upregulated in ductal adenocarcinoma of the breast. Some studies uncovered several unique roles of mucins, which include modulation in proliferative, invasive and metastatic potential of tumour cells, throughout the progression of breast cancer.

There are only a few studies that evaluated the clinical implication of *MUC*2 expression in breast cancer tissue. Walsh et al. reported that BC patients whose tumours were positive for *MUC*2 had a significantly shorter survival compared to those with *MUC*2 non-expressing tumours (49 vs. 75 months) [14]. However, Rakha et al. found that there is no significant prognostic association in survival analysis with *MUC*2 expression in BC [15]. Do et al. also reported that there is no significant prognostic association in survival analysis with *MUC*2 expression in BC regarding both overall survival (OS) and disease-free survival (DFS) [16]. In our study, serum *MUC*2 levels were not associated with survival (*p* = 0.65) (Figure 3).

Rakha et al. reported a significant inverse association between *MUC*2 expression and ER status [15]. Patel et al. found *MUC*2 negative tumours were associated with a higher histological grade [17]. On the contrary, Ohashi et al. found no clinical implication regarding *MUC*2 expression in BC [18]. We also did not notice any prognostic or clinical implication other than a possible diagnostic role of serum *MUC*2 levels.

The limitations of our study were that it was a single-centre, observational study, and other mucin levels were not evaluated. We did not consider the effects of radiotherapy, hormone therapy, or targeted therapies; these factors may have been affected our results.

## 5. Conclusions

We discovered that serum *MUC*2 levels play a diagnostic role, although we did not find any predictive and prognostic role in survival in BC patients. For use as a diagnostic or screening tool, serum levels of *MUC*2 should be studied on a larger scale.

## Figures and Tables

**Figure 1 biomolecules-09-00040-f001:**
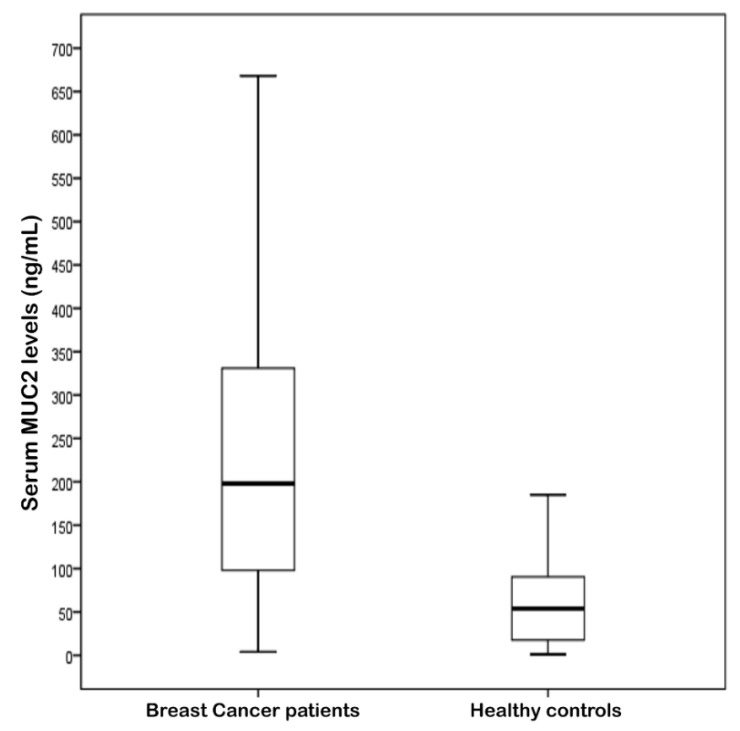
The values of serum *MUC*2 levels in patients with breast cancer and healthy controls (*p* < 0.001).

**Figure 2 biomolecules-09-00040-f002:**
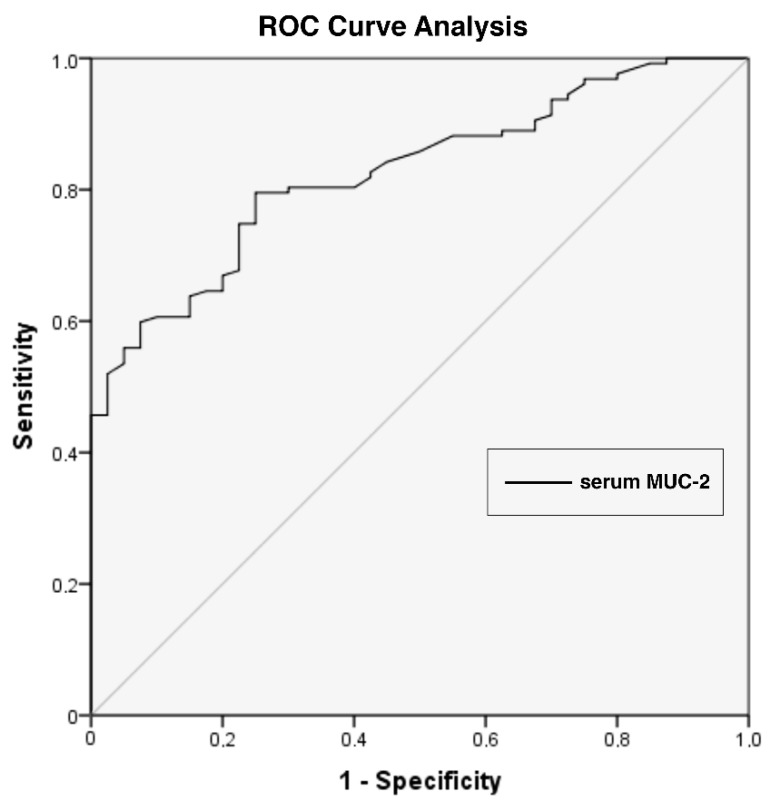
The receiver operating characteristic (ROC) analysis for serum *MUC*-2 levels (*p* < 0.001). AUC (CI) = 0.82 (0.76–0.89), cut-off value = 97.5 ng/mL, PPV= 90.5%, NPV= 48.4%. AUC: area under the curve; CI: confidence interval; PPV: positive predictive value; NPV: negative predictive value.

**Figure 3 biomolecules-09-00040-f003:**
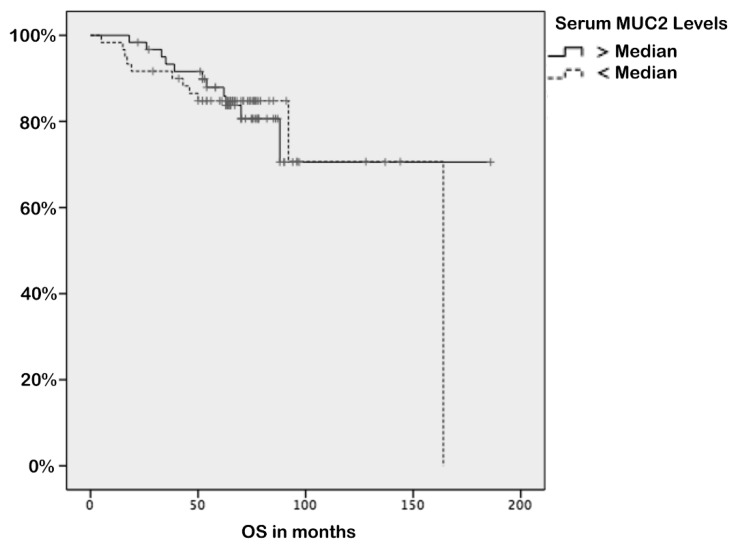
Kaplan–Meier survival curves in breast cancer patients according to serum *MUC*2 levels (*p* = 0.65). OS: overall survival.

**Table 1 biomolecules-09-00040-t001:** Characteristics of the patients and disease.

Variables	*n*
Number of patients	127
Age, years Median (range): 50 (26–78) <50/≥50	62/65
Grade 1/2/3	5/50/44
Estrogen Receptor Negative/Positive	31/96
Progesteron Receptor Negative/Positive	48/79
HER-2 Negative/Positive	97/30
classification Luminal/Triple-positive/Her-2 disease/Triple-negative	77/19/11/20
T stage * 1/2/3/4	34/58/4/3
N stage * 0/1/2/3	35/34/21/9
M stage 0/1	99/28
Serum LDH level (ULN: 450 U/L) Normal/Elevated	83/44
Serum CA15-3 level (ULN: 35 U/mL) Normal/Elevated	97/30
Response to CT Yes/No	58/7
Last status Alive/Dead	105/22

* in 99 non-metastatic patients. LDH: lactate dehydrogenase, ULN: upper limit of normal, CT: chemotherapy.

**Table 2 biomolecules-09-00040-t002:** The values of membrane-bound mucin 2 (*MUC*2) in patients with breast cancer and healthy controls.

Data	Value
Assay	MUC2 level (ng/mL)
Patients Median (range)	198 (4–468)
Controls Median (range)	54 (1–210)
*p*	<0.001 **

** *p* < 0.05.

**Table 3 biomolecules-09-00040-t003:** Results (median and range) of comparisons between the factors and various clinical/laboratory parameters.

Variables	*MUC*2 (ng/mL)	*p*
Age, years		
Younger (<50 years)	185.5 (8–640)	0.90
Older (≥50 years)	200.0 (4-668)	
Grade		
I–II (Good–medium)	215 (10–668)	0.42
III (Poor)	183.5 (4–560)	
Estrogen Receptor		
Negative	135 (4–560)	0.21
Positive	202.5 (8–668)	
Progesteron Receptor		
Negative	184 (4–668)	0.43
Positive	205 (10–560)	
HER-2		
Negative	205 (8–668)	0.51
Positive	160 (4–560)	
Classification HER-2 -negative luminal Others	205 (8–668) 160 (4–560)	0.46
Classification Triple-positive Others	183 (50–560) 201.5 (4–668)	0.73
Classification		
HER-2 disease Others	96 (4–475) 202.5 (8–668)	0.13
Classification Triple-negative Others	196 (38–560) 198 (4–668)	0.77
T stage		
1 (small)	234 (10–496)	0.41
2-4 (large)	185 (4–668)	
N status		
Negative	234 (10–490)	0.66
Positive	190.5 (4–668)	
M stage		
M0	198 (4–668)	0.65
M1	195 (20–640)	
Serum LDH level (ULN: 450 U/L)		
Normal	200 (10–668)	0.47
Elevated	175 (4–640)	
Serum CA15-3 level (ULN: 35 U/mL)		
Normal	198 (4–668)	0.84
Elevated	200.5 (14–560)	
Response to CT		
Yes No	184.5 (8–668) 270 (50–400)	0.53

LDH: lactate dehydrogenase, ULN: upper limit of normal; CT: chemotherapy.

**Table 4 biomolecules-09-00040-t004:** Univariate analyses of survival.

Variables	Mean Survival (SE) Months	3-Year Survival Rate (SE) %	*p*
Age of patients			
<50 years	84.9 (3.1)	94.7 (3.0)	0.69
≥50 years	84.8 (3.4)	89.6 (4.0)	
Grade			
I–II	89.8 (2.4)	97.4 (2.5)	0.26
III	82.0 (3.1)	91.8 (3.9)	
Estrogen Receptor			
Negative	77.7 (5.7)	82.8 (7.0)	0.03 **
Positive	87.2 (2.5)	95.3 (2.3)	
Progesteron Receptor			
Negative	80.5 (4.5)	84.8 (5.3)	0.03 **
Positive	87.0 (2.4)	97.1 (2.0)	
HER-2			
Negative	84.0 (2.8)	89.8 (3.2)	0.56
Positive	87.7 (3.8)	92.6 (5.0)	
Classification			
Luminal	86.9 (2.8)	94.3 (2.8)	0.19
Others	81.8 (4.1)	88.9 (4.7)	
Classification			
Triple-positive	83.1 (4.4)	100	0.49
Others	84.4 (2.6)	90.9 (2.9)	
Classification			
HER2 disease	86.2 (6.3)	NR	0.86
Others	84.7 (2.6)	91.4 (2.8)	
Classification Triple-negative Others	69.1 (7.5) 87.3 (2.3)	72.2 (10.6) 95.9 (2.0)	0.01 **
T stage			
T1	90.0 (0.8)	NR	0.03 **
T2–T4	84.7 (3.0)	91.7 (3.6)	
Node status			
Negative	96.8 (3.2)	87.3 (2.5)	0.24
Positive	93.2 (3.3)	85.1 (2.9)	
M stage			
M0	86.9 (2.1)	94.5 (2.4)	<0.001 **
M1	70.6 (6.6)	82.6 (7.9)	
Serum LDH level (ULN: 450 U/L)			
Normal	87.6 (2.7)	92.2 (3.1)	0.07
Elevated	77.1 (3.8)	92.3 (4.3)	
Serum CA15-3 level (ULN: 35 U/mL)			
Normal	87.5 (2.3)	95.4 (2.3)	0.004 **
Elevated	75.9 (6.0)	82.6 (7.1)	
Response to CT			
Yes	82.1 (3.8)	91.8 (3.9)	0.001 **
No	47.0 (12.4)	57.1 (18.7)	
Serum MUC2 level			
Elevated (>median)	83.7 (4.1)	89.5 (5,0)	0.65
Normal (<median)	85.2 (2.8)	93.5 (2.8)	

** *p* < 0.05. Abbreviations, LDH: lactate dehydrogenase; ULN: upper limit of normal; CT: chemotherapy.

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
