# Peer review of "Clinical Significance of Serum Membrane-Bound Mucin-2 Levels in Breast Cancer"

_biomolecules, 2019, doi:10.3390/biom9020040_

Reviewer 1 Report

Re: Clinical Significance Of Serum Membrane-Bound Mucin - 2 (Muc2) Levels In Breast Cancer

In this study, Bademler et al. investigate the serum levels of MUC2 in breast cancer and its relationship with tumour progression and prognosis. They also assess serum MUC2 levels in healthy controls. To assess the levels of MUC2 in serum they use an ELISA method.  Their data indicate that MUC2 levels are significantly higher in the serum of breast cancer patients compared to healthy controls. However, they did not find associations between serum MUC2 levels and prognosis. The work presented is novel as serum MUC2 levels had not been previously investigated in breast cancer, and although negative, results must be shared with the scientific and public community. However, it is not clear how the authors envisage the use of assessing the levels of MUC2 levels in breast cancer.

Comments:

1.    Although serum MUC2 levels are significantly higher in breast cancer patients than in healthy controls, some patients have MUC2 levels similar to controls. Yet, the authors did not find an explanation for this difference in MUC2 levels across breast cancer patients. Therefore, how sensitive and specific would be a diagnostic test based on MUC2 levels?

2.    Bademler et al. assess serum MUC2 levels in relationship with chemotherapy response and they find no association. What about other therapies such as radiotherapy, hormone therapy or targeted therapies?

3.    Do serum MUC2 levels correlate with MUC2 protein or gene expression in tumours?

4.    Please note that the classification of breast cancer as luminal and HER2-enriched is not accurate. Luminal and HER2-enriched subtype should be used only based on the molecular classification determined by PAM50 gene expression. Luminal tumours are not always HER2-negative, while HER2-enriched tumours are not always HER2-positive. This terminology could be confusing and should be modified.

5.    In table 1, M stage should be 99 for M0 and 28 for M1.

Author Response

Response to Reviewer 1 Comments

We, authors, thank you for your valuable evaluation. All your comments were found very helpful for clarity of the paper and were appreciated. The changes made in regard to your comments are listed below.

 1.     Although serum MUC2 levels are significantly higher in breast cancer patients than in healthy controls, some patients have MUC2 levels similar to controls. Yet, the authors did not find an explanation for this difference in MUC2 levels across breast cancer patients. Therefore, how sensitive and specific would be a diagnostic test based on MUC2 levels?

Answer: Added to Results. 

2.     Bademler et al. assess serum MUC2 levels in relationship with chemotherapy response and they find no association. What about other therapies such as radiotherapy, hormone therapy or targeted therapies?

Answer: Added to limitations. 

3.     Do serum MUC2 levels correlate with MUC2 protein or gene expression in tumours?

Answer: No related literature can be found. An advanced search made in Pubmed [(serum MUC2[Title/Abstract]) AND (correlate[Title/Abstract] OR correlation[Title/Abstract] OR associated[Title/Abstract]) AND (mRNA[Title/Abstract] OR gene expression[Title/Abstract])].

4.     Please note that the classification of breast cancer as luminal and HER2-enriched is not accurate. Luminal and HER2-enriched subtype should be used only based on the molecular classification determined by PAM50 gene expression. Luminal tumours are not always HER2-negative, while HER2-enriched tumours are not always HER2-positive. This terminology could be confusing and should be modified.

Answer: Thanks for your valuable comment. Problems fixed. 

5.     In table 1, M stage should be 99 for M0 and 28 for M1.

Answer: Problem fixed. 

 Yours sincerely

Reviewer 2 Report

I do not agree with the use as a survival parameter the last contact with family and I believe that if there are no concrete data about  the survival parameters these should be excluded from the study.  The configuration of tables without visible markers of cells makes it difficult to analyse them.

However  I think that it is an interesting study and that the results although not conclusive evidence the importance of continuing to study this parameter as a possible biomarker and that should be published

Author Response

Response to Reviewer 2 Comments

We, authors, thank you for your valuable evaluation. All your comments were found very helpful for clarity of the paper and were appreciated. The changes made in regard to your comments are listed below.

 1.     I do not agree with the use as a survival parameter the last contact with family and I believe that if there are no concrete data about the survival parameters these should be excluded from the study.  

Answer: Unfortunately, this suggestion is unacceptable. Firstly, we did not include the data that is not checked into the study. The dates of deaths were checked at MERNIS, an online system. Secondly, in our country, if a patient dies, a close relative of died patient informs the physician. Furthermore, during the terminal stages of a patient, his/her doctor arranges the treatment frequently. Most of the time, death is an expected end of the process. Those make our data trustworthy.

2.     The configuration of tables without visible markers of cells makes it difficult to analyse them.

Answer: Problem fixed. 

 Yours sincerely

Round  2

Reviewer 1 Report

The authors now say that "serum MUC2 levels was 60.6% and 85%, respectively, if the cut-off value was taken as 157 ng/mL (median)". They should show a ROC-curve and calculate index of true positive, false positive, true negative and false negative. These analysis are important to claim that "serum MUC2 levels play a diagnostic role". Description of the analysis should be added in the methods. 

It is unclear why the authors have not evaluated MUC2 levels in relationship with response to radiotherapy, hormone therapy or targeted therapies. Were the data not available?

Author Response

Response to Reviewer Comments

We thank you for your valuable evaluation. Your comments were found very helpful for clarity of the paper and were appreciated. The changes made in regard to your comments are listed below.

The authors now say that "serum MUC2 levels was 60.6% and 85%, respectively, if the cut-off value was taken as 157 ng/mL (median)". They should show a ROC-curve and calculate index of true positive, false positive, true negative and false negative. These analysis are important to claim that "serum MUC2 levels play a diagnostic role". Description of the analysis should be added in the methods.

Answer: A ROC analysis is conducted and reported 

It is  unclear why the authors have not evaluated MUC2 levels in relationship with response to radiotherapy, hormone therapy or targeted therapies. Were the data not available?

Answer: Although we agree that we need to make these evaluations, unfortunately, the data we have, unfortunately, were not sufficient to make these assessments.

 Kind regards,

Dr Alisan Zirtiloglu